# Connection between encounter volume and diffusivity in geophysical flows

Irina I. Rypina[1], Stefan G. Llewellyn Smith[2], and Larry J. Pratt[1]

[1]Physical Oceanography Department, Woods Hole Oceanographic Institution, 266 Woods Hole Rd., Woods Hole, MA 02543

[2]Department of Mechanical and Aerospace Engineering, Jacobs School of Engineering and Scripps Institution of Oceanography, UCSD, 9500 Gilman Dr., La Jolla, CA 92093-0411

Corresponding author's email: irypina@whoi.edu

**Abstract:** Trajectory encounter volume – the volume of fluid that passes close to a reference fluid parcel over some time interval – has been recently introduced as a measure of mixing potential of a flow. Diffusivity is the most commonly used characteristic of turbulent diffusion. We derive the analytical relationship between the encounter volume and diffusivity under the assumption of an isotropic random walk, i.e. diffusive motion, in one and two dimensions. We apply the derived formulas to produce maps of encounter volume and the corresponding diffusivity in the Gulf Stream region of the North Atlantic based on satellite altimetry, and discuss the mixing properties of Gulf Stream rings. Advantages offered by the derived formula for estimating diffusivity from oceanographic data are discussed, as well as applications to other disciplines.

1. Introduction

The frequency of close encounters between different objects or organisms can be a fundamental metric in social and mechanical systems. The chances that a person will meet a new friend or contract a new disease during the course of a day is influenced by the number of distinct individuals that he or she comes into close contact with. The chances that a predator will ingest a poisonous prey, or that a mushroom hunter will mistakenly pick up a poisonous variety, is influenced by the number of distinct species or variety of prey or mushrooms that are encountered. In fluid systems, the exchange of properties such as temperature, salinity or humidity between a given fluid element and its surroundings is influenced by the number of other distinct fluid elements that pass close by over a given time period. In all these cases it is best to think of close encounters as providing the *potential*, if not necessarily the act, of transmission of germs, toxins, heat, salinity, etc...

In cases of property exchange within continuous media such as air or water, it may be most meaningful to talk about a mass or volume passing within some radius of a reference fluid element as this element moves along its trajectory. Rypina and Pratt (2017) introduce a trajectory encounter volume, $V$, the volume of fluid that comes in contact with the reference fluid parcel over a finite time interval. The increase of $V$ over time is one measure of the mixing potential of the element, "mixing" being the irreversible exchange of properties between different water

parcels. Thus, fluid parcels that have large encounter volumes as they move through the flow
field have large mixing potential, i.e., an opportunity to exchange properties with other fluid
parcels, and vice versa.
In order to formally define the encounter volume $V$, Rypina and Pratt (2017) subdivide the entire
fluid into infinitesimal fluid elements with volumes $dV_i$, and define the encounter volume for
each fluid element to be the total volume of fluid that passes within a radius $R$ of it over a finite
time interval $t_0 < t < t_0 + T$, i.e.,
$$V(\vec{x}_0; t_0, T, R) = \lim_{dV_i \to 0} \Sigma_i \, dV_i. \tag{1}$$
In practice, for dense uniform grids of trajectories, $\overrightarrow{x_k}(\vec{x}_{0k}; t_0, T), k = 1, \dots, K$, where $t_0$ is the
starting time, $T$ is the trajectory integration time, and $\vec{x}_{0k}$ is the trajectory initial position
satisfying $\vec{x}(\vec{x}_0, t_0; T = 0) = \vec{x}_0$, both the limit and the subscript in the above definition (1) can
be dropped. In this case, the encounter volume can be approximated by
$$V \approx N \, \delta V, \tag{2}$$
where the *encounter number*,
$$N(\vec{x}_{0ref}; t_0, T, R) = \sum_{\substack{k=1 \\ k \neq l}}^{K} \mathrm{I}\left(\min\left(\left|\overrightarrow{x_k}(\vec{x}_{0k}; t_0, T) - \vec{x}_{ref}(\vec{x}_{0ref}; t_0, T)\right|\right) \leq R\right), \tag{3}$$
is the number of trajectories that come within a radius $R$ of the reference trajectory,
$\vec{x}_{ref}(\vec{x}_{0ref}; t_0, T)$, over a time $t_0 < t < t_0 + T$. Here the indicator function I is 1 if true and 0 if
false, and $K$ is the total number of particles. As in Rypina and Pratt (2017), we define encounter
volume based on the number of encounters with different trajectories, not the total number of
encounter events (see the schematic diagram of trajectory encounters in Fig. 1). Rypina and Pratt
(2017) discuss how the encounter volume can be used to identify Lagrangian Coherent
Structures (LCS) such as stable and unstable manifolds of hyperbolic trajectories and regions
foliated by the KAM-like tori surrounding elliptic trajectories in realistic geophysical flows. A
detailed comparison between the encounter volume method and some other Lagrangian methods
of LCS identification, as well as the dependences on parameters, $t_0, T, R$, and on grid spacing (or
on the number of trajectories, K), and the relative advantages of different techniques, was given
in Rypina and Pratt (2017). The interested reader is referred to that earlier paper for details. The
current paper is concerned only with the question of finding the connection between the
encounter volume and diffusivity, rather than identifying LCS.
Given the seemingly fundamental importance of close encounters, it is of interest to relate
metrics such as $V$ to other bulk measures of interactions within the system. For example, in some
cases it may be more feasible to count encounters rather than to measure interactions or property
exchanges directly, whereas in other cases the number of encounters might be most pertinent to
the process in question but difficult to measure directly. In many applications, including ocean
turbulence, the most commonly used metric of mixing is the eddy diffusivity, $\kappa$, a quantity that
relates transport of fluid elements by turbulent eddies to diffusion (LaCasce, 2008; Vallis, 2006;
Rypina et al., 2015; Kamenkovich et al., 2015). The underlying assumption is that the eddy field
drives downgradient tracer transfer, similar to molecular diffusion but with a different (larger)
diffusion coefficient. This diffusive parameterization of eddies has been implemented in many
non-eddy-resolving oceanic numerical models. The diffusivity can be measured by a variety of
means, including dye release (Ledwell et al., 2000; Sundermeyer and Ledwell, 2001; Rypina et
al., 2016), surface drifter dispersion (Okubo, 1971; Davis, 1991; LaCasce, 2008, La Casce et al.,
2014; Rypina et al. 2012; 2016), and property budgets (Munk, 1966). In numerical models $\kappa$ is
often assumed constant in both time and space, or related in some simplified manner to the large-
scale flow properties (Visbeck, 1997).
Because the purpose of the diffusivity coefficient $\kappa$ is to quantify the intensity of the eddy-
induced tracer transfer, i.e., the intensity of mixing, it is tempting to relate it to the encounter
volume, $V$, which quantifies the mixing potential of a flow and thus is closely related to tracer
mixing. Such an analytical connection between the encounter volume and diffusivity could
potentially also be useful for the parameterizations of eddy effects in numerical models. The
main goal of this paper is to develop a relationship between $V$ and $\kappa$ in one and two dimensions.
Specifically, we seek an analytical expression for the encounter volume, $V$, i.e., the volume of
fluid that passed within radius $R$ from a reference particle over time, as a function of $\kappa$. The
relationship is not as straightforward as one might first imagine, but can nevertheless be written
down straightforwardly in the long-time limit. This is opportune, since the concept of eddy
diffusivity is most relevant in the long-time limit.
2.   Connection between encounter volume and diffusivity
This problem was framed in mathematical terms in Rypina and Pratt (2017), who outlined some
initial steps towards deriving the analytical connection between encounter volume and diffusivity
but did not finish the derivation. In this section, we complete the derivation.
2.1. Main idea for the derivation
Let us start by considering the simplest diffusive random walk process in one or two dimensions,
where particles take steps of fixed length $\Delta x$ in random directions along the x-axis in 1D or
along both x- and y-axes in 2D, respectively, at fixed time intervals $\Delta t$.
The single particle dispersion, i.e., the ensemble-averaged square displacement from the
particle's initial position, is $D_{1D} = <(x - x_0)^2>$ and $D_{2D} = <(x - x_0)^2 + (y - y_0)^2>$ in 1D
or 2D, respectively. For a diffusive process, the dispersion grows linearly with time, and the
constant proportionality coefficient related to diffusivity. Specifically, $D_{1D} = 2\kappa_{1D}t$ with
$\kappa_{1D} = \Delta x^2 / (2\Delta t)$, and $D_{2D} = 4K_{2D}t$ with $\kappa_{2D} = \Delta x^2 / (4\Delta t)$.
It is convenient to consider the motion in a reference frame that is moving with the reference
particle. In that reference frame, the reference particle will always stay at the origin, while other
particles will still be involved in a random walk motion, but with a diffusivity twice that in the
stationary frame, $\kappa^{moving}{=}2\kappa^{stationary}$ (Rypina and Pratt, 2017).
The problem of finding the encounter number then reduces to counting the number of randomly
walking particles (with diffusivity $\kappa^{moving}$) that come within radius $R$ of the origin in the
moving frame. This is related to a classic problem in statistics – the problem of a random walker
reaching an absorbing boundary, usually referred to as "a cliff" (because once a walker reaches
the absorbing boundary, it falls off the cliff), over a time interval from 0 to $t$.
In the next section we will provide formal solutions; here we simply outline the steps to
streamline the derivation. We start by deriving the appropriate diffusion equation for the
probability density function, $p(\vec{x}, t)$, of random walkers in 1D or 2D:
$$\frac{\partial p}{\partial t} = \kappa \nabla^2 p. \tag{4}$$
We place a cliff, $\vec{x_c}$, at the perimeter of the encounter sphere, i.e., at a distance $R$ from the
origin, and impose an absorbing boundary condition at a cliff,
$$p(\vec{x_c}, t) = 0, \tag{5a}$$
which removes (or "absorbs") particles that have reached the cliff (see Fig. 2 for a schematic
diagram). We then consider a random walker that is initially located at a point $\vec{x_0}$ outside the
cliff at $t = 0$, i.e.,
$$p(\vec{x}, t = 0) = \delta(\vec{x} - \vec{x_0}), \tag{5b}$$
and we write an analytical solution for the probability density function satisfying Eqs. (4-5),
$$G(\vec{x}, t; \vec{x_0}, \vec{x_c}), \tag{6}$$
that quantifies the probability to find a random walker initially located at $\vec{x_0}$ at any location $\vec{x}$
outside of the cliff at a later time $t > 0$. In mathematical terms, $G$ is the Green's function of the
diffusion equation.
The survival probability, which quantifies the probability that a random walker initially located
at $\vec{x_0}$ at $t = 0$ has "survived" over time $t$ without falling off the cliff, is
$$S(t; \vec{x_0}, \vec{x_c}) = \int G(\vec{x}, t; \vec{x_0}, \vec{x_c}) d\vec{x}, \tag{7}$$
where the integral is taken over all locations outside of the cliff. The encounter, or "non-
survival", probability can then be written as the conjugate quantity,
$P_{en}(t; \overrightarrow{x_0}, \overrightarrow{x_c}) = 1 - S(t; \overrightarrow{x_0}, \overrightarrow{x_c}),$ (8)
which quantifies the probability that a random walker initially located at $\overrightarrow{x_0}$ at $t = 0$ has
reached, or fallen off, the cliff over time $t$. This allows one to write the encounter volume, i.e.,
the volume occupied by particles that were initially located outside of the cliff and that have
reached the cliff by time $t$, as
$V(t; \overrightarrow{x_c}) = \int P_{en}(t; \overrightarrow{x_0}, \overrightarrow{x_c}) d\overrightarrow{x_0},$ (9)
where the integral is taken over all initial positions outside of the cliff.
2.2. 1D case
Consider a random walker initially located at the origin, who takes, with a probability ½, a fixed
step $\Delta x$ to the right or to the left along the x-axis after each time interval $\Delta t$. Then the probability
to find a walker at a location $x = n\Delta x$ at after $(m + 1)$ steps is
$p(n\Delta x, (m + 1)\Delta t) = 1/2[p((n - 1)\Delta x, m\Delta t) + p((n + 1)\Delta x, m\Delta t)].$ (10)
Using a Taylor series expansion in $\Delta x$ and $\Delta t$, we can write down the finite-difference
approximation to the above expression as

$$p(x,t) + \Delta t \frac{\partial p}{\partial t} = \frac{1}{2}\left[p(x,t) - \Delta x \frac{\partial p}{\partial x} + \frac{\Delta x^2}{2}\frac{\partial^2 p}{\partial x^2} + p(x,t) + \Delta x \frac{\partial p}{\partial x} + \frac{\Delta x^2}{2}\frac{\partial^2 p}{\partial x^2} + O(\Delta x^4)\right] =$$

$= p(x,t) + \frac{\Delta x^2}{2}\frac{\partial^2 p}{\partial x^2} + O(\Delta x^4),$ (11)
yielding a diffusion equation
$\frac{\partial p}{\partial t} = \kappa \frac{\partial^2 p}{\partial x^2}$ (12)
with diffusivity coefficient $\kappa = \frac{\Delta x^2}{2\Delta t}$.

A Green's function for the 1D diffusion equation without a cliff is a solution with initial
condition $p(x, t = 0; x_0) = \delta(x - x_0)$ in an unbounded domain. It takes the form
$G_{unbounded}(x, t; x_0) = \frac{1}{\sqrt{4\pi\kappa t}} e^{-\frac{(x-x_0)^2}{4\kappa t}}.$ (13)
A Green's function with the cliff (see Fig. 2 for a schematic diagram), for a solution to the
initial-value problem with $p(x, t = 0; x_0) = \delta(x - x_0)$ in a semi-infinite domain, $x \in [-\infty; x_c]$,
with an absorbing boundary condition at a cliff, $p(x = x_c, t; x_0) = 0$, can be constructed by the
method of images from two unbounded Green's functions as

$$G(x, t; x_0, x_c) = \frac{1}{\sqrt{4\pi\kappa t}}(e^{-\frac{(x-x_0)^2}{4\kappa t}} - e^{-\frac{(x-(2x_c-x_0))^2}{4\kappa t}}). \tag{14}$$

It follows from (7-9) that the survival or non-encounter probability is

$$S(t; x_0, x_c) := \int_{-\infty}^{x_c} G(x, t; x_0, x_c)dx = Erf[\frac{x_c-x_0}{2\sqrt{\kappa t}}], \tag{15}$$

the encounter probability is

$$P_{en}(t; x_0, x_c) = 1 - S(t) = 1 - Erf\left(\frac{x_c-x_0}{2\sqrt{\kappa t}}\right), \tag{16}$$

and the encounter volume is

$$V(t; x_c) = \int_{-\infty}^{x_c} P_{en}(t; x_0, x_c)dx_0 = \int_{-\infty}^{x_c}\left(1 - Erf\left[\frac{x_c-x_0}{2\sqrt{\kappa t}}\right]\right)dx_0 = \frac{2}{\sqrt{\pi}}\sqrt{\kappa t}. \tag{17}$$

The above formula accounts for the randomly walking particles that have reached the cliff from
the left over time $t$. By symmetry, if the cliff was located to the right of the origin, the same
number of particles would be reaching the cliff from the right, so the total encounter volume is

$$V(t; x_c) = \frac{4}{\sqrt{\pi}}\sqrt{\kappa t}. \tag{18}$$

Note that formula (18) gives the encounter volume, i.e., the volume of fluid coming within radius
$R$ from the origin, in a reference frame moving with the reference particle, so the corresponding
diffusivity in the right-hand side of (18) is $\kappa^{moving} = 2\kappa^{stationary}$.
2.3. 2D case
Consider a random walker in 2D, who is initially located at the origin and who takes, with a
probability of $1/4$, a fixed step of length $\Delta x$ to the right, left, up or down after each time interval
$\Delta t$. Then the probability to find a walker at a location $x = n\Delta x, y = m\Delta x$ at time $t = (m + 1)\Delta t$
is

$$p(n\Delta x, (m + 1)\Delta t) = 1/4[p((n - 1)\Delta x, m\Delta y, l\Delta t) + p((n + 1)\Delta x, m\Delta y, l\Delta t) +$$
$$+p(n\Delta x, (m - 1)\Delta y, l\Delta t) + p(n\Delta x, (m + 1)\Delta y, l\Delta t)]. \tag{19}$$
Using a Taylor series expansion in $\Delta x$, $\Delta y$ and $\Delta t$, the finite-difference approximation leads to a
diffusion equation

$$\frac{\partial p}{\partial t} = \kappa\left(\frac{\partial^2 p}{\partial x^2} + \frac{\partial^2 p}{\partial y^2}\right) \tag{20}$$

with diffusivity coefficient $\kappa = \frac{\Delta x^2}{4\Delta t}$.
To proceed, we need an analytical expression for the Green's function of Eq. (20) with a cliff at a
distance $R$ from the origin, i.e., a solution to the initial-value problem with $p(\vec{x}, t = 0; \vec{x_0}) =$
$\delta(\vec{x} - \vec{x_0})$ for the above 2D diffusion equation on a semi-infinite plane ($r \geq R, 0 < \theta \leq 2\pi$),
bounded internally by an absorbing boundary (a cliff) located at $r = R$, so that $p(r =$
$R, \theta, t; \vec{x_0}) = 0$ (see Fig. 2(right) for a schematic diagram). Here $(r, \theta)$ are polar coordinates.
Carlslaw and Joeger (1939) give the answer as
$$G(r, \theta, t; r_0, \theta_0, R) = u + w = \sum_{n=-\infty}^{\infty}(u_n(r, t; r_0, R) + w_n(r, t; r_0, R)) \cos n(\theta - \theta_0) \qquad (21)$$
where $r_0(\geq R), \theta_0$ denote the source location, and
$$\{u_n, w_n\} = L^{-1}\left\{\bar{u}_n, \bar{w}_n\right\} = \frac{1}{2\pi i}\lim_{T\to\infty}\int_{\gamma-iT}^{\gamma+iT} e^{st}\left\{\bar{u}_n, \bar{w}_n\right\}ds$$
are the inverse Laplace transforms of
$$\bar{u}_n = \frac{1}{2\pi\kappa}\begin{cases}I_n(qr)K_n(qr_0), R < r < r_0 \\ I_n(qr_0)K_n(qr), r > r_0\end{cases} \text{ and } \bar{w}_n = -\frac{I_n(qR)}{K_n(qR)}K_n(qr_0)K_n(qr) \qquad (22)$$
with $q = \sqrt{\frac{s}{\kappa}}$.
The survival probability (from Eq. (7)) is
$$S(t; r_0, R) = \int_{R^2} G(\vec{x}, t; \vec{x_0}, R)d^2\vec{x} = \int_0^{2\pi}\int_R^{\infty}\sum_{n=-\infty}^{\infty}(u_n + v_n) \cos n(\theta - \theta_0)\, r\, dr\, d\theta =$$
$$2\pi \int_R^{\infty}(u_0 + v_0)\, r\, dr. \qquad (23)$$
Next, we take the Laplace transform of the survival probability and write it in terms of a Laplace
variable $s$ as
$$\bar{S}(s, r_0, R) = \int_0^{\infty} e^{-st}S(t; r_0, R)dt = 2\pi\int_R^{\infty}(\overline{u_0} + \overline{w_0})\, r\, dr = \frac{1}{\kappa}\int_R^{r_0} I_0(qr)K_0(qr_0)\, r\, dr +$$
$$\frac{1}{\kappa}\int_{r_0}^{\infty} I_0(qr_0)K_0(qr)\, r\, dr - \frac{1}{\kappa}\int_R^{\infty}\frac{I_0(qR)}{K_0(qR)}K_0(qr)K_0(qr_0)\, r\, dr. \qquad (24)$$
Using $\int rI_0(r)dr = rI_1(r)$ and $\int rK_0(r)dr = -rK_1(r)$, and $\lim_{x\to\infty} xK_1(x) = 0$ we find

$\bar{S}(s; r_0, R) =$
$$\frac{1}{\kappa}K_0(qr_0)\left[\frac{r}{q}I_1(qr)\right]\Big|_R^{r'} + \frac{1}{\kappa}I_0(qr_0)\left[-\frac{r}{q}K_1(qr)\right]\Big|_R^{\infty} - \frac{1}{\kappa}\frac{I_0(qr_0)}{K_0(qR)}K_0(qr_0)\left[-\frac{r}{q}K_1(qr)\right]\Big|_R^{\infty} =$$
$$\frac{1}{\kappa}\left\{\frac{r_0}{q}\left(I_1(qr_0)K_0(qr_0) + I_0(qr_0)K_1(qr_0)\right) - \frac{a}{q}\frac{K_0(qr_0)}{K_0(qR)}\left(I_1(qR)K_0(qR) + I_0(qR)K_1(qR)\right)\right\}. \qquad (25)$$
But $I_1(x)K_0(x) + I_0(x)K_1(x) = \frac{1}{x}$ so
$\overline{S}(s; r_0, R) = \frac{1}{\kappa}\left(\frac{1}{q^2} - \frac{1}{q^2}\frac{K_0(qr_0)}{K_0(qR)}\right) = \frac{1}{s}\left(1 - \frac{K_0(qr_0)}{K_0(qR)}\right).$ (26)
From (8), the encounter probability $P_{en}(t; \overrightarrow{x_0}, R) = 1 - S(t; \overrightarrow{x_0}, R)$, and from (9) the encounter
volume is
$V(t; R) = \int_{R^2} P_{en} d^2\overrightarrow{x_0} = \int_0^{2\pi}\int_R^\infty P_{en}\, r_0\, dr_0 = 2\pi\int_R^\infty [1 - S(t; r_0, R)]r_0\, dr_0.$ (27)
We now take the Laplace transform of the encounter number to get
$\overline{V}(s; R) = \int_0^\infty e^{-st}V(t; R)dt = 2\pi\int_R^\infty\left[\frac{1}{s} - \overline{S}(s; R)\right]r_0\, dr_0 = 2\pi\int_R^\infty \frac{K_0(qr_0)}{K_0(qR)}\frac{r_0}{s}\, dr_0 =$
$\frac{2\pi}{sK_0(qR)}\left[-\frac{r_0}{q}K_1(qr_0)\right]\Big|_R^\infty = \frac{2\pi R}{sq}\frac{K_1(qR)}{K_0(qR)} = \frac{2\pi R}{s^{3/2}\,\kappa^{-\frac{1}{2}}}\frac{K_1\left(\sqrt{\frac{s}{\kappa}}R\right)}{K_0\left(\sqrt{\frac{s}{\kappa}}R\right)},$ (28)
where we used $\int_0^\infty e^{-st}dt = \frac{1}{s}$, $\int K_0(z)z\, dz = -zK_1(z)$, and $\lim_{z\to\infty} K_1(z)=0$.
The explicit connection between the encounter volume and diffusivity is thus given by the
inverse Laplace transform of the above expression (28),
$V(t; R) = L^{-1}\{\overline{V}(s; R)\}.$ (29)
Although numerically straightforward to evaluate, a non-integral analytic form does not exist for
this inverse Laplace transform. To better understand the connection between $V$ and $\kappa$ and the
growth of $V$ with time, we next look at the asymptotic limits of small and large time. The small-$t$
limit is transparent, while the long-$t$ limit is more involved.

226       (a) small-$t$ asymptotics

In the small-$t$ limit, the corresponding Laplace coordinate $s$ is large, giving
$\overline{V}(s; R) \sim 2\pi R\kappa^{\frac{1}{2}}\frac{1}{s^{3/2}}$ (30)
because $lim_{z\to\infty} \frac{K_1(z)}{K_0(z)} = 1$. Noting that $L^{-1}\left\{s^{-\frac{3}{2}}\right\} = \frac{2\sqrt{t}}{\sqrt{\pi}}$, the inverse Laplace transform of the
above gives the following simple expression connecting the encounter number and diffusivity at
short times:
$V(t; R) \xrightarrow{t\to 0} 4R\sqrt{\pi}\,\sqrt{\kappa t}.$ (31)

233       (b) large-$t$ asymptotics

In the large-$t$ limit, the Laplace coordinate $s$ is small and the asymptotic expansions $K_0, K_1$ take
the form
$$lim_{z\to 0} K_0(z) = -\gamma - \ln\left(\frac{z}{2}\right) + O\left(\left(\frac{z}{2}\right)^2 \ln\left(\frac{z}{2}\right)\right), \tag{32}$$
$$lim_{z\to 0} K_1(z) = \frac{1}{z} + \frac{z}{2}\left[\ln\left(\frac{z}{2}\right) + \gamma - \frac{1}{2}\right] + O\left(z^3 \ln z\right), \tag{33}$$
giving
$$lim_{s\to 0} \bar{V}(s; R) = -\frac{4\pi\kappa}{s^2 \ln(\tau s)} - \frac{\pi R^2}{s} + O\left(\frac{1}{s\ln(\tau s)}\right), \tag{34}$$
where
$$\tau = \frac{R^2 e^{2\gamma}}{4\kappa}. \tag{35}$$
We now need to take an inverse Laplace transform of $\bar{V}$. The second term in the right-hand side
gives $L^{-1}\left\{\frac{\pi R^2}{s}\right\} = \pi R^2$. Llewelyn Smith (2000) discusses the literature for inverse Laplace
transforms of the form $(s^\alpha \ln s)^{-1}$ for small $s$. For our problem, the discussion in Olver (1974,
Chap. 8, §11.4) is the most helpful approach. His result (11.13), discarding the exponential term
which is not needed here, shows that the inverse Laplace transform of $(s^2 \ln s)^{-1}$ has the
asymptotic expansion
$$L^{-1}\left\{\frac{1}{s^2 \ln s}\right\} \xrightarrow{t\to\infty} -t\left(\frac{1}{\ln t} + \frac{1-\gamma}{(\ln t)^2} + O((\ln t)^{-3})\right). \tag{36}$$
Using $L^{-1}\{F(\tau s)\} = \frac{1}{\tau} f(t/\tau)$, we thus obtain the desired connection between the encounter
number and diffusivity at long times:
$$V(t; R) \xrightarrow{t\to\infty} 4\pi\kappa t\left(\frac{1}{\ln\frac{t}{\tau}} + \frac{1-\gamma}{\left(\ln\frac{t}{\tau}\right)^2}\right) - \pi R^2 + O\left(\frac{t}{\left(\ln\frac{t}{\tau}\right)^3}\right) + O\left(\frac{1}{\ln\frac{t}{\tau}}\right). \tag{37}$$
Physically, the time scale $\tau$ (Eq. (35)) defines the time at which the dispersion of random
particles, $D = 4\kappa\tau$, is comparable to the volume of the encounter sphere, ie., $R^2 e^{2\gamma} \cong \pi R^2$ in
2D. Thus for $t \gg \tau$, particles are coming to the encounter sphere "from far away."
For practical applications, it is sufficient to only keep the leading order term of the expansion,
yielding a simpler connection between encounter number and diffusivity,
$$V(t; R) \xrightarrow{t\to\infty} \frac{4\pi\kappa t}{\ln\frac{t}{\tau}} + O\left(\frac{t}{\left(\ln\frac{t}{\tau}\right)^2}\right). \tag{38}$$
Note again that the diffusivity in the right-hand side of Eqs. (28-29), (31) and (38) is
$\kappa^{moving} = 2\kappa^{stationary}$.
2.4. Numerical tests of the derived formulas in 1d and 2d
Before applying our results to the realistic oceanic flow, we numerically tested the accuracy of
the derived formulas in idealized settings by numerically simulating a random walk motion in 1D
and 2D, as described in the beginning of subsections 2.1 and 2.2, respectively. We then
computed the encounter number and encounter volume using definition (2-3), and compared the
result with the derived exact formulas (18) and (28-29) and with the asymptotic formulas (31)
and (38). Note that although formulas (28-29) are exact, the inverse Laplace transform still needs
to be evaluated numerically and thus is subject to numerical accuracy, round-off errors etc.; these
numerical errors are, however, small, and we will refer to numerical solutions of (28-29) as
"exact," as opposed to the asymptotic solutions (31) and (38).
The comparison between numerical simulations and theory is shown in Fig. 3. Because the
numerically simulated random walk deviates significantly from the diffusive regime over short
(< O(100Δt)) time scales, the agreement between numerical simulation and theory is poor at
those times in both 1D and 2D. Once the random walkers have executed > 100 time steps,
however, the dispersion reaches the diffusive regime, and the agreement between the theory (red)
and numerical simulation (black) rapidly improves for both 1D and 2D cases, with the two
curves approaching each other at long times. In 2D, the long-time asymptotic formula (38) works
well at long times, $t \gg \tau$, as expected. The 2D short-time asymptotic formula (green) agrees well
with the exact formula (red) at short times but not with the numerical simulations (black) for the
same reason as discussed above, i.e., because the numerically simulated random walk has not yet
reached the diffusive regime at those times.
3.   Application to the altimetric velocities in the Gulf Stream region
Sea surface height measurements made from altimetric satellites provide nearly global estimates
of geostrophic currents throughout the World Oceans. These velocity fields, previously
distributed by AVISO, are now available from the Copernicus Marine and Environment
Monitoring Service (CMEMS) website (http://marine.copernicus.eu/), both along satellite tracks
and as a gridded mapped product in both near-real and delayed time. Here we use the delayed-
time gridded maps of absolute geostrophic velocities with ¼ deg spatial resolution and temporal
step of 1 day, and focus our attention on the Gulf Stream extension region of the North Atlantic
Ocean. There, the Gulf Stream separates from the coast and starts to meander, shedding cold-
and warm-core Gulf Stream rings from its southern and northern flanks. These rings are among
the strongest mesoscale eddies in the ocean. However, their coherence, interaction with each
other and with other flow features, and their contribution to transport, stirring and mixing are still
not completely understood (Bower et al., 1985; Cherian and Brink, 2016).

Maps showing the encounter volume for fluid parcel trajectories in the region, and the corresponding diffusivity estimates (Fig. 4) could be useful both for understanding and interpreting the transport properties of the flow, as well as for benchmarking and parameterization of eddy effects in numerical models. In our numerical simulations, trajectories were released on a regular grid with $dx = dy \cong 10$ km on 11 Jan 2015 and were integrated forward in time for 90 days using a fifth-order variable-step Runge-Kutta integration scheme with bi-linear interpolation between grid points in space and time. The encounter radius was chosen to be $R = 30$ km in both zonal and meridional directions, i.e., about a third of a radius of a typical Gulf Stream ring. Similar parameter values were used in Rypina and Pratt (2017), although our new simulation was carried out using more recent 2015 velocities instead of 1997 as in that paper.

The encounter volume field, shown in the top left panel of Fig. 4, highlights the overall complexity of the flow and identifies a variety of features with different mixing potential, most notably several Gulf Stream rings with spatially small low-V (blue) cores and larger high-$V$ (red) peripheries. Although the azimuthal velocities and vorticity-to-strain ratio are large within the rings, the coherent core regions with inhibited mixing potential are small, suggesting that the *coherent* transport by these rings might be smaller than anticipated from the Eulerian diagnostics such as the Okubo-Weiss or closed-streamline criteria (Chelton et al., 2011; Abernathey and Haller, 2017). On the other hand, the rings' peripheries, where the mixing potential is elevated compared to the surrounding fluid, cover a larger geographical area than the cores. Thus, while rings inhibit mixing within their small cores, the enhanced mixing on the periphery might be their dominant effect. This is consistent with the results from Rypina and Pratt (2017), but a more thorough analysis is needed to test this hypothesis. Notably, the encounter number is also large along the northern and southern flank of the Gulf Stream jet, with two separate red curves running parallel to each other and a valley in between (although the curves could not be traced continuously throughout the entire region). This enhanced mixing on both flanks of the Gulf Stream Extension current is reminiscent of chaotic advection driven by the tangled stable and unstable manifolds at the sides of the jet (del-Castillo-Negrete and Morrison, 1993; Rogerson et al., 1999; Rypina et al., 2007; Rypina and Pratt, 2017), and is also consistent with the existence of critical layers (Kuo, 1949; Ngan and Sheppard, 1997).

We now apply the asymptotic formula (38) to convert the encounter volume to diffusivity. Because equation (38) is not invertible analytically, we converted $V$ to $\kappa$ numerically using a look-up table approach. More specifically, we used (38) to compute theoretically-predicted $V$ values at time T=90 days for a wide range of $\kappa$'s spanning all possible oceanographic values for 0 to $10^9 \ cm^2/s$, and we used the resulting look-up table to assign the corresponding $\kappa$ values to $V$ values in the 3$^{rd}$ row of Fig. 4. Note that, instead of the long-time asymptotic formula (38) (as in in the 3$^{rd}$ row of Fig. 4), it is also possible to use the exact formulas (28-29) to convert $V$ to $\kappa$ via a table look-up approach. The resulting exact diffusivities, shown in the 2$^{nd}$ row of Fig. 4, are similar to the long-time asymptotic values (3$^{rd}$ row). Because both exact and asymptotic formulas

were derived under the assumption of a diffusive random walk, neither should work well in
regions with a non-diffusive behavior. The asymptotic formula has the advantage of being
simpler and it also provides for a numerical estimate of the "long-time-limit" time scale, $\tau$,
shown in the bottom row of Fig. 4
As expected, the diffusivity maps in the $2^{nd}$ and $3^{rd}$ rows of Fig. 4, which resulted from
converting $V$ to $\kappa$ using (28-29) or (38), respectively, have the same spatial variability as the $V$-
map, with large $\kappa$ at the peripheries of the Gulf Stream rings and at the flanks of the Gulf Stream
and small $\kappa$ at the cores of the rings, near the Gulf Stream centerline and far away from the Gulf
Stream current, where the flow is generally slower. The diffusivity values range from
$O(10^5)\ cm^2/s$ to $O(10^7)\ cm^2/s$. Using the 1971 Okubo's diffusivity diagram and scaling law,
$\kappa_{Okubo}[cm\^2/s] = 0.0103\ l[cm]^{1.15}$, our diffusivity values correspond to spatial scales from
$10\ km$ to $650\ km$, thus spanning the entire mesoscale range. This is not surprising considering
the Lagrangian nature of our analysis, where trajectories inside the small ($< 50\ km$) low-
diffusion eddy cores stay within the cores for the entire integration duration (90 days), whereas
trajectories in the high-diffusivity regions near the ring peripheries and at the flanks of the Gulf
Stream jet cover large distances, sometimes $> 650\ km$, over 90 days.
The performances of the exact and asymptotic diffusive formulas vary greatly throughout the
domain, with better/poorer performances in high-/low-$V$ areas. This is because in the low-$V$
areas, the behavior of fluid parcels is non-diffusive, so the diffusive theoretical formulas work
poorly. The breakdown of the long-time asymptotic formula is evident in the $4^{th}$ row of Fig. 4,
which shows the corresponding long-time scales, $\tau$ (from Eq. (35)), throughout the domain. As
suggested by our 2D random walk simulations, the long-time asymptotic diffusive formula only
works well when $t \gg \tau$, but in reality $\tau$ values are $< 9$ days (1/10 of our integration time) only in
the highest-$V$ regions, and are much larger everywhere else, reaching values of $\cong 90$ days within
the cores of the Gulf Stream rings. More detailed comparison between theory, both exact and
asymptotic, and numerical $V(t)$ is shown in Fig. 5 for 3 reference trajectories that are initially
located inside the core, on the periphery, and outside of a Gulf Stream ring (black, red, and blue,
respectively). Clearly, the diffusive theory works poorly for the trajectory inside the eddy core
(black curve). The agreement is better for the blue and even better for the red curves,
corresponding to trajectories outside and on the periphery of the eddy, although deviations
between the theory and numerics are still visible, raising questions about the general validity of
the diffusive approximation in ocean flows on time scales of a few months.
The non-diffusive nature of the parcel motion over 90 days is because ocean eddies have finite
length- and time-scales, so a variety of different transport regimes generally occurs before
separating parcels become uncorrelated and transport becomes diffusive, as in a random walk. At
very short times the motion of fluid parcels is largely governed by the local velocity shear, so the
resulting transport regime is ballistic, i.e., $D \propto T^2$ and $V \propto T$ (Rypina and Pratt, 2017). At longer

times, when velocity shear can no longer be assumed constant in space and time, the regime may transition to a local Richardson regime (i.e., $D \propto t^3$), where separation at a given scale is governed by the local features of a comparable scale (Richardson 1926; Bennett 1984; Beron-Vera and LaCasce 2016), or to a non-local chaotic-advection spreading regime (i.e., $D \propto \exp(\lambda t)$), where separation is governed by the large scale flow features (Bennett 1984; Rypina et al. 2010; Beron-Vera and LaCasce 2016). The kinetic energy spectrum of a flow indicates whether a local or non-local regime will be relevant. The chaotic transport regime is generally expected to occur in mesoscale-dominated eddying flows, such as, for example, AVISO velocity fields, over time scales of a few eddy winding times. At times long enough for particles to sample many different flow features, such as Gulf Stream meanders or mesoscale eddies in the AVISO fields, the velocities of the neighboring particles become completely uncorrelated, and transport finally approaches the diffusive regime. With the mesoscale eddy turnover time being on the order of several weeks, it often takes longer than 90 days to reach the diffusive regime.

A number of diffusivity estimates other than Okubo's have been made for the Gulf Stream extension region (e.g., Zhurbas and Oh, 2004; LaCasce, 2008; Rypina et al., 2012; Abernathey and Marshall, 2013; Klocker and Abernathey, 2014; or Cole et al., 2015). These estimates are based on surface drifters (Zhurbas and Oh, 2004, LaCasce, 2008; Rypina et al., 2012), satellite-observed velocity fields (Abernathey and Marshall, 2013; Klocker and Abernathey, 2014, Rypina et al., 2012), and Argo float observations (Cole et al., 2015), and they use either the spread of drifters or the evolution of simulated or observed tracer fields to deduce diffusivity. The resulting diffusivities are spatially varying and span 2 orders of magnitude, from $2 \times 10^4$ $m^2/s$ in the most energetic regions in the immediate vicinity of the Gulf Stream and its extension, to $10^3 \ m^2/s$ in less energetic areas, to $200 \ m^2/s$ in the coastal areas of the Slope Sea. Diffusivity estimates vary significantly depending on the initial tracer distribution used (Abernathey and Marshall, 2013) and depend on whether the suppression by the mean current has been taken into account (Klocker and Abernathey, 2014). The diffusivity tensor has also been shown to be anisotropic, with a large anisotropy ratio near the Gulf Stream (Rypina et al., 2012). Data resolution and coverage, as well as the choice of time and length scales also play a role in defining $\kappa$ value (Cole at al., 2015). All of these issues complicate the reconciliation of different diffusivity estimates. Nevertheless, ignoring these complications for a moment, and avoiding the smallest diffusivities in those geographical areas of Fig. 4 where the diffusive approximation is invalid, our $O(10^3 \ m^2/s)$ encounter-volume-based diffusivity estimates tend to be in the middle of the range of available estimates for the western North Atlantic. Although not inconsistent with other estimates, the encounter volume method did not predict diffusivities to reach values of $10^4 \ m^2/s$ anywhere within the considered geographical domain.

Because the action of the real ocean velocity field on drifters or tracers is generally not exactly diffusive, all methods simply fit the diffusive approximation to the corresponding variable of interest, such as particle dispersion, tracer variance, or, in our case, encounter volume. The

analytic form of the diffusive approximation is, however, different for different variables and
different flow regimes. For example, for a diffusive random walk regime, dispersion grows
linearly with time, whereas the growth of the encounter volume is non-linear, as defined by eq.
(38). This generally leads to different diffusivity estimates resulting from different methods. In
other words, the diffusivity value that fits best to the observed particle dispersion at 90 days does
not necessarily provide the best fit to the observed encounter volume at 90 days, and vice versa.
To illustrate this more rigorously, we consider a linear strain flow,

$$u = \alpha\, x,$$

$$v = -\alpha\, y,$$

with a constant strain coefficient $\alpha$. The particle trajectories are given by $x = x_0 e^{\alpha t}, y = y_0 e^{-\alpha t}$
where $x_0, y_0$ are particles initial positions. The dispersion of a small cluster of particles that are
initially uniformly distributed within a small square of side length $2dx$ is

$$D = <(X - \bar{X})^2 + (Y - \bar{Y})^2>,$$

where $X = x - x_0$ and $Y = y - y_0$ are displacements of particles from their initial positions and
the overbar denotes the ensemble mean. Since the linear strain velocity remains unchanged in a
reference frame moving with a particle, without loss of generality we can restrict our attention to
a cluster that is initially centered at the origin, so $\bar{X} = \bar{Y} = 0$. In the long time limit, when
$e^{\alpha t} \gg 1 \gg e^{-\alpha t}$, the dispersion becomes

$$D = 2/3 dx^3 e^{2\alpha t}.$$

If one is using a diffusive fit,

$$D = 4\kappa_D t,$$

to approximate diffusivity, then the resulting diffusivity is

$$\kappa_D = \frac{dx^3 e^{2\alpha t}}{6t}.$$

On the other hand, the encounter volume for the linear strain flow is

$$V = 2\alpha R^2 t,$$

whereas the long-time diffusive fit is

$$V = \frac{4\pi \kappa_V t}{\ln t/\tau},$$

yielding

$$\kappa_V = -\frac{\alpha R^2 ProductLog(-\frac{\pi e^{2\gamma}}{2\alpha t})}{2\pi}$$

where the function $ProductLog(z)$ is a solution to $z = we^w$. Because $\kappa_D$ is exponential in time,
while $\kappa_V$ is not, $\kappa_D$ always becomes larger than $\kappa_V$ at large $t$.
Of course, real oceanic flows are more complex than the simple linear strain example. However,
for flows that are in a state of chaotic advection, exponential separation between neighboring
particles will occur and the dispersion will grow exponentially in time, as in the linear strain
example. Although we do not have a formula for the encounter volume for a chaotic advection
regime, the linear strain example suggests that the encounter volume growth will likely be slower
than exponential. Thus, for a chaotic advection regime, the dispersion-based diffusivity could be
expected to be larger than the encounter-volume-based diffusivity. This can potentially explain
the smaller encounter-volume-based diffusivity values in Fig. 4 compared to other available
estimates from the literature. Numerical simulations (not shown) using an analytic Duffing
oscillator flow, which features chaotic advection, indeed produced smaller encounter-volume-
based diffusivity than dispersion-based diffusivity, in agreement with our arguments above. The
AVISO velocities are dominated by the meso- rather than submeso-scales, and the 90-day time
interval is about a few mesoscale eddy winding times, thus this flow satisfies all the pre-
requisites for the chaotic advection to occur. Finally, the particle trajectories that we used to
produce Fig. 4 can be grouped into small clusters (we are using encounter radius R=30 km as a
cluster radius for consistency) to estimate their dispersion and infer diffusivity from its slope.
Consistent with our arguments above, the resulting dispersion-based diffusivities in Fig. 6 are
larger than the encounter-volume-based diffusivities in Fig. 4 and reach values $O(10^4 \ m^2/s)$ in
the energetic regions of the Gulf Stream and its extension, in agreement with the previous
diffusivity estimates from the literature. In applications where the number of encounters is a
more important quantity than the spread of particles, the encounter-volume-based diffusivity
might be a more appropriate estimate to use.
In the left panels of Fig. 4 we used the full velocity field to advect trajectories, so both the mean
and the eddies contributed to the resulting encounter volumes and the corresponding
diffusivities. But what is the contribution of the eddy field alone to this process? To answer this
question, we have performed an additional simulation in the spirit of Rypina et al. (2012), where
we advected trajectories using the altimetric time-mean velocity field, and then subtracted the
resulting encounter volume, $V_{mean}$, from the full encounter number, $V$. The result characterizes
the contribution of eddies, although strictly speaking $V_{eddy} \neq V - V_{mean}$ because of non-
linearity. Note also that because we are interested in the Lagrangian-averaged effects of eddies
following fluid parcels, $V_{eddy}$ cannot be estimated by simply advecting particles by the local
eddy field alone (see an extended discussion of this effect in Rypina et al., 2012). Not
surprisingly, the eddy-induced encounter volumes (upper right panel of Fig. 4) are smaller than
the full encounter numbers, with the largest decrease near the Gulf Stream current, where both
the mean velocity and the mean shear are large. In other geographical areas, specifically at the
peripheries of the Gulf Stream rings, the decrease in $V$ is less significant, so the resulting map
retains its overall qualitative spatial structure. The same is true for the diffusivities in the 2nd and
$3^{rd}$ rows of Fig. 4. The overall spatial structure of the eddy diffusivity is preserved and matches
that in left panels, but the values decrease, with the largest differences near the Gulf Stream,
where some diffusivity values are now $O(10^6) \ cm^2/s$ instead of $O(10^7) \ cm^2/s$. In contrast, $\kappa$
only decreases, on average, by a factor of 2 (instead of an order of magnitude) near the
peripheries of the Gulf Stream rings. The long-time diffusive time scale $\tau$ generally increases,
and the ratio $t/\tau$ generally decreases throughout the domain, but the long-time asymptotic
formula (38) still works well in high-$V$ regions, specifically on the peripheries of the Gulf Stream
rings where $\tau$ is still significantly less than $t$.
4. Discussion and Summary
With many new diagnostics being developed for characterizing mixing in fluid flows, it is
important to connect them to the well-established conventional techniques. This paper is
concerned with understanding the connection between the encounter volume, which quantifies
the mixing potential of the flow, and diffusivity, which quantifies the intensity of the down-
gradient transfer of properties. Intuitively, both quantities characterize mixing and it is natural to
expect a relationship between them, at least in some limiting sense. Here, we derived this
anticipated connection for a diffusive process, and we showed how this connection can be used
to produce maps of spatially-varying diffusivity, and to gain new insights into the mixing
properties of eddies and the particle spreading regime in realistic oceanic flows.
When applied to the altimetry-based velocities in the Gulf Stream region, the encounter volume
and diffusivity maps show a number of interesting physical phenomena related to transport and
mixing. Of particular interest are the transport properties of the Gulf Stream rings. The
materially-coherent Lagrangian cores of these rings, characterized by very small diffusivity, are
smaller than expected from earlier Eulerian diagnostics (Chelton et al., 2011). The periphery
regions with enhanced diffusivity are, on the other hand, large, raising a question about whether
the rings, on average, act to preserve coherent blobs of water properties or to speed up the
mixing. The encounter volume, through the derived connection to diffusivity, might provide a
way to address this question and to quantify the two effects, clarifying the role of eddies in
transport and mixing.
Our encounter-volume-based diffusivity estimates are within the range of other available
estimates from the literature, but are not among the highest. We provided an intuitive explanation
for why the encounter-volume-based diffusivities might be smaller than the dispersion-based
diffusivities, and we supported our explanation with theoretical developments based on a linear
strain flow, and with numerical simulations. We note that in problems where the encounters
between particles are of interest, rather than the particle spreading, the encounter-volume-based
diffusivities would be more appropriate to use than the conventional dispersion-based estimates.
Reliable data-based estimates of eddy diffusivity are needed for parameterizations in numerical
models. The conventional estimation of diffusivity from Lagrangian trajectories via calculating
particle dispersion requires large numbers of drifters or floats (LaCasce, 2008). It would be
useful to have a technique that would work with fewer instruments. The derived connection
between encounter volume and diffusivity might help in achieving this goal. Specifically, one
could imagine that if an individual drifting buoy was equipped with an instrument that would
measure its encounter volume – the volume of fluid that came in contact with the buoy over time
t – then the resulting encounter volume could be converted to diffusivity using the derived
connection. This would allow estimating diffusivity using a single instrument.
In the field of social encounters, it is becoming possible to construct large data sets by tracking
cell phones, smart transit cards (Sun, et al. 2013), and bank notes (Brockmann, et al. 2006). As
was the case for the Gulf Stream trajectories, some of the behavior appears to be diffusive and
some not so. Where diffusive/random walk behavior is relevant, it may be easier to accumulate
data on close encounters rather than on other metrics using, for example, autonomous vehicles
and instruments that are able, through local detection capability, to count foreign objects that
come within a certain range.
**Acknowledgments:** This work was supported by the NSF grants OCE-1558806 and EAR-
1520825, and NASA grant NNX14AH29G.

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

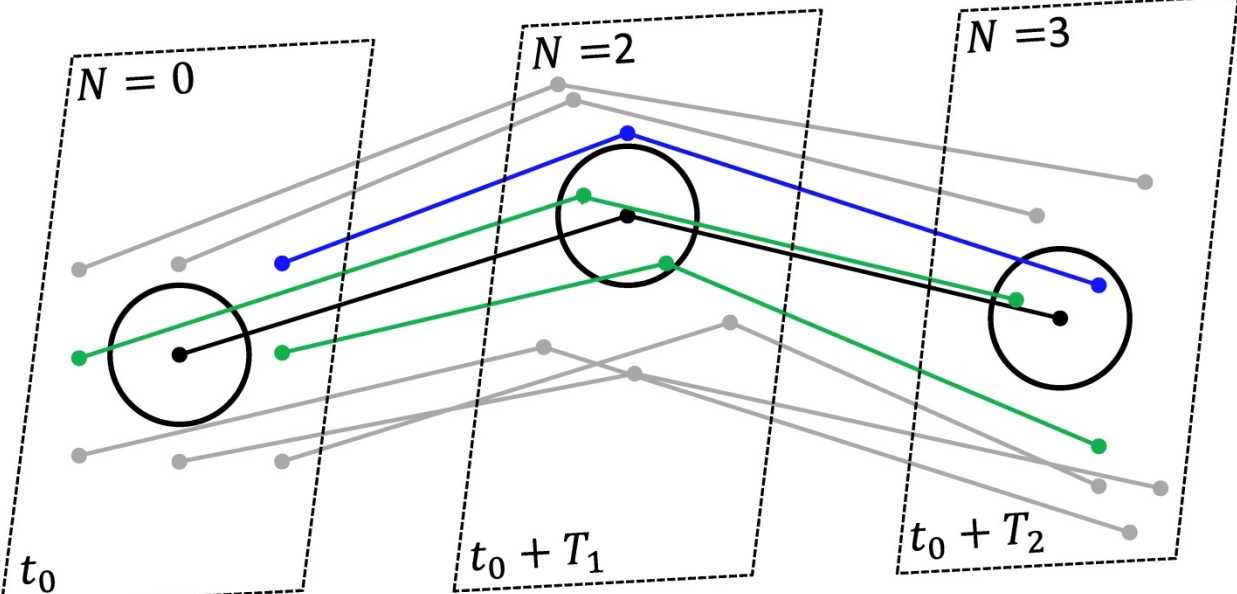

Fig. 1. Schematic diagram of trajectory encounters, showing trajectories of 9 particles, with dots indicating positions of particles at 3 time instances, at the release time, $t_0$, and at two later times, $t_0 + T_1$ and $t_0 + T_2$. The reference trajectory and the encounter sphere are shown in black, trajectories that do not encounter the reference trajectory are in grey, and trajectories that encounter the reference trajectory are in green if the encounter occur at $t_0 + T_1$, and in blue if encounters occur at $t_0 + T_2$. Time slices are schematically shown by dashed rectangles, and the encounter number, $N$, is indicated at the top of each time slice.

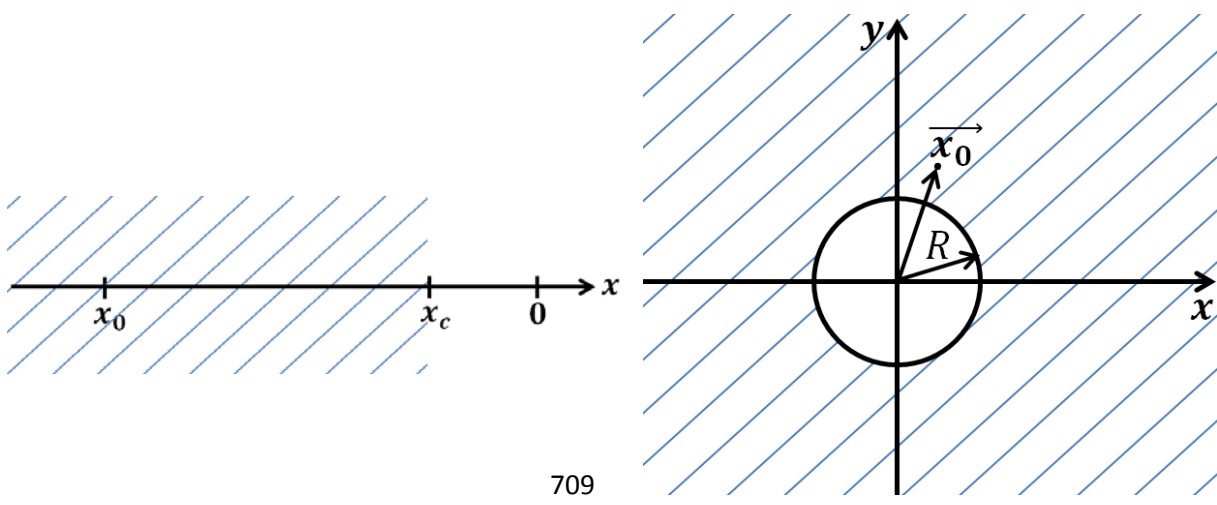


**Figure 2. Schematic diagram in 1D (left) and 2D (right). Hatched areas show semi-infinite domains outside of the cliff.**










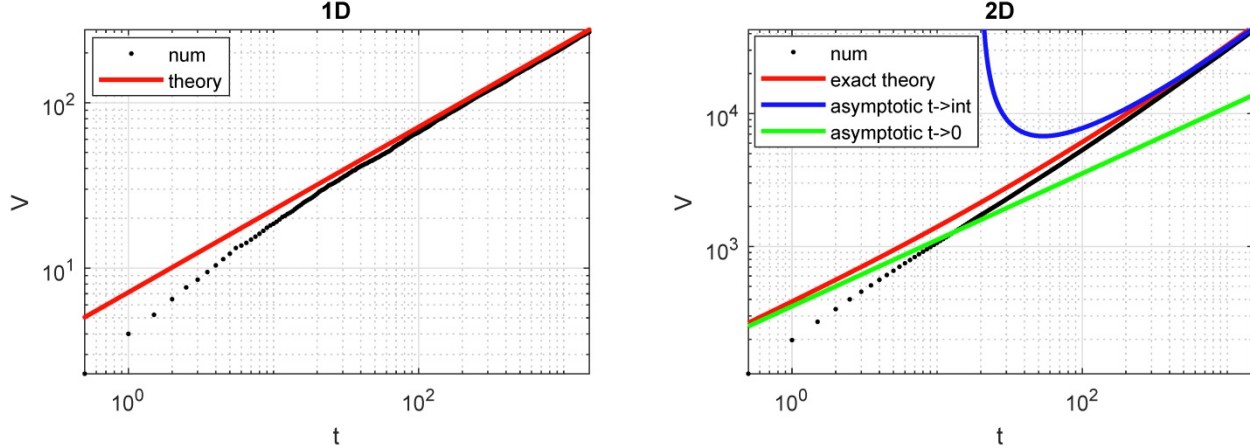


**Figure 3. Comparison between theoretical expression (red, green, blue) and numerical estimates (black) of the encounter**
**volume for a random walk in 1D (left) and 2D (right). In both, $\kappa = 5$ and $\Delta t = 0.5$. In 2D, $\tau \cong 20$.**




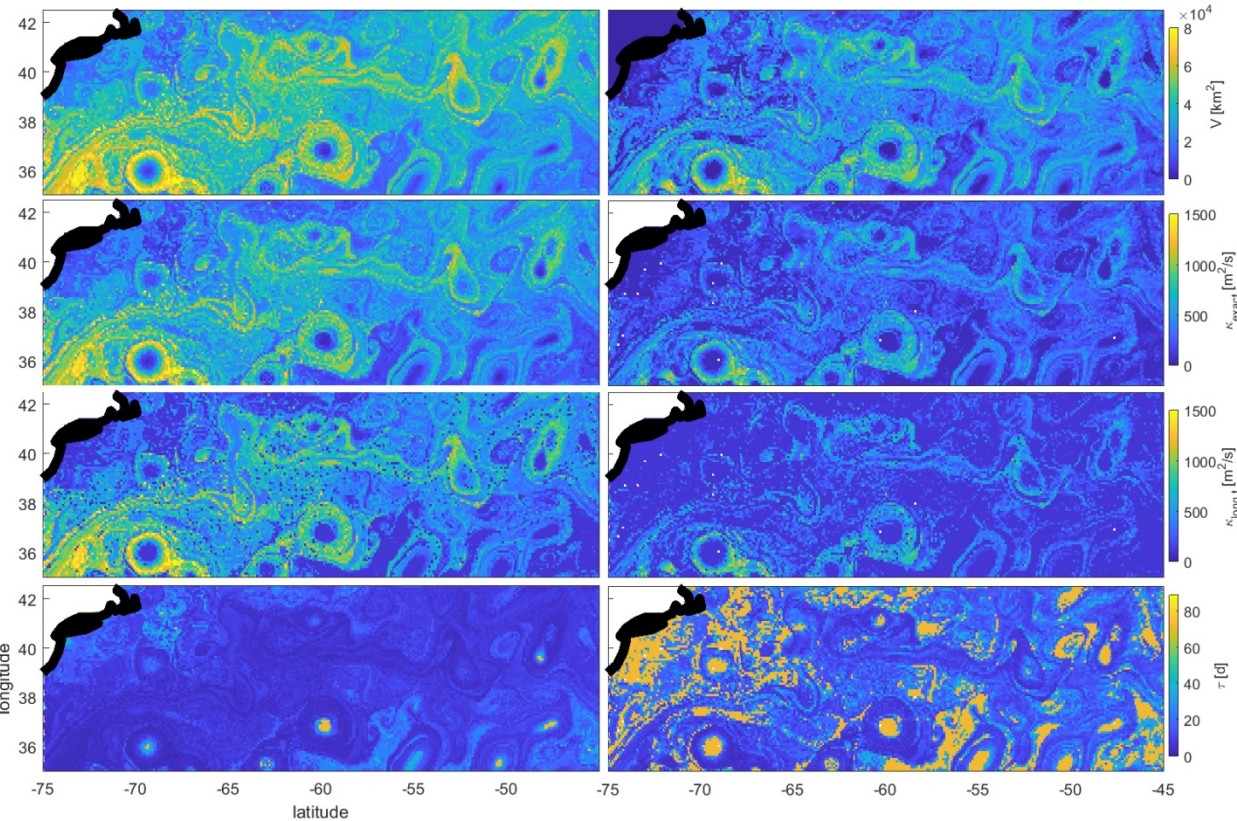


**Figure 4. (will add 4 right panels corresponding to eddy-induced estimates) Encounter number (1st row), exact diffusivity**
**(2nd row), long-time diffusivity (3rd row) and diffusive time-scale (4-th row) for the full flow (left) and for the eddy**
**component of the flow (right). The encounter volume was computed on 11/01/2015 over 90 days with a radius of 3°. The**
**lower panel shows comparison between numerically-computed $V$ and the long-time diffusive formula (38) with the**
**corresponding $\kappa$ for the 3 reference trajectories located in the core, periphery and outside (blue, magenta, green) of the**
**Gulf Stream ring.**



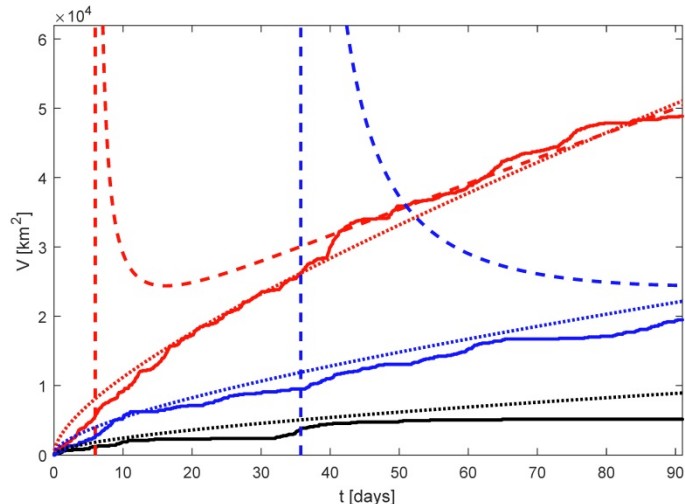


**Figure 5. Comparison between numerically-computed** $V$ **(solid) and the exact (dotted) and long-time diffusive formulas (dashed) with the corresponding** $\kappa$ **for the 3 reference trajectories located in the core, periphery and outside (black, red, blue) of the Gulf Stream ring.**












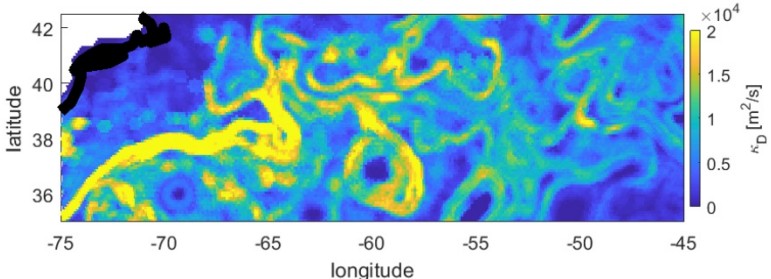


**Figure 6. Dispersion-based diffusivity, $\kappa_D$.**