# Peer review of "Connection between encounter volume and diffusivity in geophysical flows"

_Nonlinear Processes in Geophysics, 2017_

## Referee Comment (RC1) · Anonymous Referee #1 · 20 Nov 2017

This article presents a Lagrangian technique to estimate or measure the diffussion from the "volume encounter" magnitude defined by the authors as the volume of fluid that passes close to a reference fluid parcel over some time interval. The authors derive the analitical connection between this magnitude and the diffusion coefficient for 1d and 2d problems. Then, using the equations derived for the 2d problem, they apply it to ocean data from altimetry to reproduce and obtain maps of diffusivity for eddies on the Gulf Stream region.

This technique may have a strong potential to stimate in a smart way the diffusivity from Lagrangian trajectories data. However, the paper needs major revisions before

continuing the reviewing process,

1. The article is heavy deducing equations but short in results. The resuls section should be increased. A quantitative comparison of the diffusivity obtained through this method could be compared with other previous methods or results.

2. There are no cites to other problems related with the close encounters. The authors should cite preceding work where this concept was used to introduce the reader.

3. The article should be reestructured. In my opinion the article has many subdivisions which don't make easy to follow it. Once the "context" is given at the first paragraph, there is an informal definition of the volume encounter V (paragraph 2). Then, on the third paragraph the authors introduce the problem, then the methodology used. Then, again on the section (1.1) the authors introduce formal definitions, then describe in detail the problem again on the section (1.2). I suggest the authors, the description of the problem should be done at once, then present the hipothesis and finally the methodology that are going to be used to address the problem.

4. The notation should be improved to make it clear. There are undefined symbols. Some examples: -. Line 64. There is no definition for x. -. x_k is the trajectory k and x are the points which start at position xâĆĂ and time tâĆĂ and they have been integrated for a time T? The definition of the trajectories should be improved to make it clear. -. Line 103: "take steps of fixed length L". If it is an step the authors should be coherent with the notation an use $\delta L$ -. Line 118: "over a time inteval t". "t" is not an interval. Maybe you mean t \in[tâĆĂ,T]

5. It would be helpfull the use of schematic images to show what is an encounter volume, with the corresponding notation. As an example, in Haller (2016) there are many schematic figures to introduce the reader on the concept and the notation.

6. The citation of the different Lagrangian methods is quite awkard. The citations are used just to mention properties of the Lagrangian problem itself. They should motivate

better the use of these citations.

7. There are severals sentences related to the choices or the scales of the problem that should be rewriten or at least support it with some reference.

8. In line 69 is written "The encounter volume depends on the starting time, integration time, the number of trajectories and the encounter radious. The dependences on the first three parameters are typical for all Lagrangian methods." Does the authors' method offer some advantage or diference higher than the mentioned? Methods as FTLE or FSLE depends on those three properites but other properties are derived from the particles trajectories. The LAVD also uses the vorticity along the trajectory. The authors should clarify better what is the advange or the difference than its method has over those ones.

9. line 74. "The integration time should be long enough for trajectories to sample the features of interest well, but short enough compared to their lifetime." What happens if the time required to "sample of feature" is similar to lifetime? The second part of this sentence should be deleted or clarified. This is very arbitrary and depend so much of the phenomena in question.

10. "The grid spacing should be smaller compared to the sizes of the features of interest, and the encounter radios should be smaller than about half of the size of the smallest features of interest." The idea behind this is to ensure that your technique will capture the smallest scales on the flow given mainly by the flow resolution. On this paragraph there are some assumptions that should be make it clear.

11. The image quality is really poor and the details cannot be appreciated.

---

## Referee Comment (RC2) · Anonymous Referee #2 · 26 Nov 2017

The manuscript presents an extension of a previous work (Rypina and Pratt, 2017), relating the 'trajectory encounter volume' , V, defined there to the 'effective diffusivity' in an ocean flow.

The manuscript presents interesting developments that would merit publication in NPG. Nevertheless, there are a few points the authors should revise before I can recommend publication:

- The authors compute V by assuming trajectories behave as random walks, which is the main assumption behind the concept of 'effective diffusivity' , k. This gives an expression (in 2d only implicit) for V in terms of k. Then V is computed in a real flow, and

the formula is used to plot k. There is a step to close the logic here, and it is to compare the k obtained in this way with the one obtained in other approaches aiming also at identifying k from data. I understand that this may be difficult, in particular because, even as shown in this paper, the 'effective diffusivity' concept is far from appropriate in many situations, and in those in which it works, only at very long times. But in my opinion something should be said al least for the regions in which the diffusivity approach seems consistent. If not a full comparison with other results, at least some discussion beyond the simple consistency with the diffusivities of Okubo (1971) for mesoscale spatial scales.

- The random walk theory is developed for a circular (in 2d) area around the focus trajectory. But in the application to the ocean the authors consider a region with R= 0.3 degrees. This is something like a rectangle or trapezoid (on the sphere) and its size in kilometers will change when the focal trajectory changes latitude. Could you explain more clearly which is the region you actually use when computing numerically V, and its relationship with the circular region of the theory?

- In several places of the paper a 'fit' of the data to Eq. 38 is mentioned. Please state more clearly if this fit is only used to asses the validity of the diffusivity assumptions or if it is even used to estimate k (the most direct estimation, instead, would not use the temporal fit, but just the value of V at time T).

- The description of figure 3 in lines 363-367 seems to have some error (red and black curves are mentioned which do not appear in the figure, and also the mentioned order 'core, periphery and outside' should be probably 'core, outside and periphery'.

- The relative motion of two random walks is just another random walk of double diffusivity. But for fluid trajectories, absolute dispersion and relative dispersion may be quite different. In fact there is a most interesting regime of 'nonlocal transport' for relative dispersion before the asymptotic diffusion regime is reached. I think something should be commented on thee different time regimes in relation with the very long times needed

to observe consistently the diffusive regime for relative dispersion.

- Please use consistently either \delta x or L in the developments of Sect. 2.

- The quality of the figures is quite low, which is specially important for Fig. 3.
* * *

---

## Author Comment (AC1) · 30 Jan 2018

This article presents a Lagrangian technique to estimate or measure the diffusion from the "volume encounter" magnitude defined by the authors as the volume of fluid that passes close to a reference fluid parcel over some time interval. The authors derive the analytical connection between this magnitude and the diffusion coefficient for 1d

and 2d problems. Then, using the equations derived for the 2d problem, they apply it to ocean data from altimetry to reproduce and obtain maps of diffusivity for eddies on the Gulf Stream region.

This technique may have a strong potential to estimate in a smart way the diffusivity from Lagrangian trajectories data. However, the paper needs major revisions before continuing the reviewing process.

We thank the reviewer for a positive view on our paper and for important comments and suggestions.

R1-1. The article is heavy deducing equations but short in results. The results section should be increased. A quantitative comparison of the diffusivity obtained through this method could be compared with other previous methods or results.

We consider the theoretical developments to be one of the important results of this paper, and we tried to explain the derivation of the analytical connection between encounter volume and diffusivity in enough detail for a reader to follow through. Thus, Sec 2 ended up being somewhat lengthy. However, we feel that shortening this section would make it less straightforward for a reader to follow, and for this reason we decided not to shorten it in the revision.

Instead, following the reviewer's comment, in the revised paper we have extended Sec. 3 to include a comparison of our encounter-volume-based diffusivity estimates with other available diffusivity estimates from the existing literature, including estimates by Zhurbas and Oh, 2004; LaCasce, 2008; Rypina et al., 2012; Abernathey and Marshall, 2013; Clocker and Abernathey, 2014; and Cole et al., 2015.Our diffusivity estimates are within the range of available estimates, but not among the highest. We provided an intuitive explanation for why the encounter-volume-based diffusivities may be smaller than the dispersion-based diffusivities, and we supported our explanation with theoretical developments for a linear strain flow and with additional numerical simulations. We note that in problems where the encounters between particles are of interest, rather

than the particle spreading, encounter-volume-based diffusivities would be more appropriate to use than conventional dispersion-based estimates. Please see the extended discussion of this issue on pp. 13-15 and 17 of the revised paper.

R1-2. There are no cites to other problems related with the close encounters. The authors should cite preceding work where this concept was used to introduce the reader.

We are not aware of any physical oceanography papers (except for Rypina and Pratt, 2017), which use the concept of encounters. In other disciplines, we have tracked down a few references that made use of this concept, and we have included these references to the last paragraph of the Conclusions section of the revised paper. A more exhaustive review of encounter-related literature across all areas would of course be interesting, but is beyond the scope of the current paper.

R1-3. The article should be restructured. In my opinion the article has many subdivisions which don't make easy to follow it. Once the "context" is given at the first paragraph, there is an informal definition of the volume encounter V (paragraph 2). Then, on the third paragraph the authors introduce the problem, then the methodology used. Then, again on the section (1.1) the authors introduce formal definitions, then describe in detail the problem again on the section (1.2). I suggest the authors, the description of the problem should be done at once, then present the hypothesis and finally the methodology that are going to be used to address the problem.

Following the reviewer's comment, we have restructured the revised paper. Specifically, we removed all subsections from the Introduction, and significantly streamlined the text.

R1-4. The notation should be improved to make it clear. There are undefined symbols. Some examples: - Line 64. There is no definition for x. -. $x_k$ is the trajectory k and x are the points which start at position xâ′C ËŸA and time tâ′C ËŸA and they have been integrated for a time T? The definition of the trajectories should be improved to make it clear. - Line 103: "take steps of fixed length L". If it is an step the authors should be coherent with the notation an use $\delta L$ -. Line 118: "over a time inteval t". "t" is

not an interval. Maybe you mean t \in[tâ′CAËŸ ,T]

In the revised paper, the notation has been clarified and undefined symbols have been defined.

R1-5. It would be helpfull the use of schematic images to show what is an encounter volume, with the corresponding notation. As an example, in Haller (2016) there are many schematic figures to introduce the reader on the concept and the notation.

We have included a schematic diagram (new Fig. 1) showing trajectory encounters.

R1-6. The citation of the different Lagrangian methods is quite awkard. The citations are used just to mention properties of the Lagrangian problem itself. They should motivate better the use of these citations.

We agree that for the purposes of this paper, which focuses on a connection between encounter volume and diffusivity, other Lagrangian methods are not directly relevant because they focus on a different question - the question of identifying Lagrangian coherent structures (rather than connection to diffusivity). Thus, we have removed these references from the revision. The new text on p. 2 now reads: "Rypina and Pratt (2017) discuss how the encounter volume can be used to identify Lagrangian Coherent Structures (LCS) such as stable and unstable manifolds of hyperbolic trajectories and regions foliated by the KAM-like tori surrounding elliptic trajectories in realistic geophysical flows. A detailed comparison between the encounter volume method and some other Lagrangian methods of LCS identification, as well as the dependences on parameters, $t_0, T, R$, and on grid spacing (or on the number of trajectories, $K$), and the relative advantages of different techniques, was given in Rypina and Pratt (2017). The interested reader is referred to that earlier paper for details. The current paper is concerned only with the question of finding the connection between the encounter volume and diffusivity, rather than identifying LCS."

R1-7. There are several sentences related to the choices or the scales of the problem

that should be rewritten or at least support it with some reference.

We have deleted most of the paragraph in question. Please see our answer to R1-6 and R1-8 for more detail.

R1-8. In line 69 is written "The encounter volume depends on the starting time, integration time, the number of trajectories and the encounter radious. The dependences on the first three parameters are typical for all Lagrangian methods." Does the authors' method offer some advantage or difference higher than the mentioned? Methods as FTLE or FSLE depends on those three properties but other properties are derived from the particles trajectories. The LAVD also uses the vorticity along the trajectory. The authors should clarify better what is the advantage or the difference that this method has over those ones.

A detailed comparison between the encounter volume method and other Lagrangian methods of LCS identification, as well as the dependences on parameters and on grid spacing, and the relative advantages of different techniques, was given in Rypina et al., 2017. We feel that it would not be appropriate to repeat this discussion here, as it would only distract from the main focus of this paper, which is the connection between the encounter volume and diffusivity (rather than comparison between, and the relative pros and cons of, different LCS identification methods). For this reason, we have removed most of the paragraph in question from the revised paper. The interested reader is referred to Rypina et al., 2017.

R1-9. line 74. "The integration time should be long enough for trajectories to sample the features of interest well, but short enough compared to their lifetime." What happens if the time required to "sample of feature" is similar to lifetime? The second part of this sentence should be deleted or clarified. This is very arbitrary and depend so much of the phenomena in question.

We have deleted the sentence in question, as the reviewer suggested. Please see also our answer to R1-6 and R1-8.

R1-10. "The grid spacing should be smaller compared to the sizes of the features of interest, and the encounter radius should be smaller than about half of the size of the smallest features of interest." The idea behind this is to ensure that your technique will capture the smallest scales on the flow given mainly by the flow resolution. In this paragraph there are some assumptions that should be made it clear.

We have deleted the sentence in question. Please see also our responces to R1-6 and R1-8.

R1-11. The image quality is really poor and the details cannot be appreciated.

The image quality has been significantly improved in the revised manuscript.

Anonymous Referee #2

The manuscript presents an extension of a previous work (Rypina and Pratt, 2017), relating the 'trajectory encounter volume', V, defined there to the 'effective diffusivity' in an ocean flow. The manuscript presents interesting developments that would merit publication in NPG. Nevertheless, there are a few points the authors should revise before I can recommend publication.

We thank the reviewer for a positive view on our paper and for important comments and suggestions.

R2-1. The authors compute V by assuming trajectories behave as random walks, which is the main assumption behind the concept of 'effective diffusivity' , k. This gives an expression (in 2d only implicit) for V in terms of k. Then V is computed in a real flow, and the formula is used to plot k. There is a step to close the logic here, and it is to compare the k obtained in this way with the one obtained in other approaches aiming also at identifying k from data. I understand that this may be difficult, in particular because, even as shown in this paper, the 'effective diffusivity' concept is far from appropriate in many situations, and in those in which it works, only at very long times. But in my opinion something should be said at least for the regions in which the diffusivity

approach seems consistent. If not a full comparison with other results, at least some discussion beyond the simple consistency with the diffusivities of Okubo (1971) for mesoscale spatial scales.

Following the reviewer's comment, in the revised paper we have significantly extended Sec. 3 to include a comparison of our encounter-volume-based diffusivity estimates with other available diffusivity estimates from the existing literature, including Zhurbas and Oh, 2004; LaCasce, 2008; Rypina et al., 2012; Abernathey and Marshall, 2013; Clocker and Abernathey, 2014; and Cole et al., 2015. Our diffusivity estimates are within the range of available estimates, but not among the highest. We provided an intuitive explanation for why the encounter-volume-based diffusivities may be smaller than the dispersion-based diffusivities, and we have supported our explanation with theoretical developments for a linear strain flow and with additional numerical simulations. We note that in problems where the encounters between particles are of interest, rather than the particle spreading, encounter-volume-based diffusivities would be more appropriate to use than conventional dispersion-based estimates. Please see the extended discussion of this issue on pp. 13-15 and 17 of the revised paper.

R2-2. The random walk theory is developed for a circular (in 2d) area around the focus trajectory. But in the application to the ocean the authors consider a region with R= 0.3 degrees. This is something like a rectangle or trapezoid (on the sphere) and its size in kilometers will change when the focal trajectory changes latitude. Could you explain more clearly which is the region you actually use when computing numerically V, and its relationship with the circular region of the theory?

We agree with the reviewer that defining the encounter radius in degrees might introduce some unnecessary confusion. To avoid this issue altogether, in the revised paper we have redone all the calculations using encounter radius in km instead of degrees (we used R=30 km because 30km is ∼=.3deg at 40N). This change didn't lead to any significant changes in the results.

R2-3. In several places of the paper a 'fit' of the data to Eq. 38 is mentioned. Please state more clearly if this fit is only used to assess the validity of the diffusivity assumptions or if it is even used to estimate k (the most direct estimation, instead, would not use the temporal fit, but just the value of V at time T).

The reviewer is correct. We used the value of V at time T, not the best fit over all times from 0 to T, to estimate kappa. We have clarified this as follows on p. 12 of the revised manuscript: "More specifically, we used (38) to compute theoretically-predicted V values at time T=90 days for a wide range of $\kappa$'s spanning all possible oceanographic values for 0 to ãĂŰ10ãĂŮˆ9 cmˆ2/s, and we used the resulting look-up table to assign the corresponding $\kappa$ values to V values in the 3rd row of Fig. 4."

We also agree with the reviewer that the word "fit" was confusing, and we use "performance" instead in the revised version: "The performances of the exact and asymptotic diffusive formulas vary greatly throughout the domain, with better/poorer performances in high-/low-V areas."

R2-4. The description of figure 3 in lines 363-367 seems to have some error (red and black curves are mentioned which do not appear in the figure, and also the mentioned order 'core, periphery and outside' should be probably 'core, outside and periphery'.

This has been corrected in the revised manuscript.

R2-5. The relative motion of two random walks is just another random walk of double diffusivity. But for fluid trajectories, absolute dispersion and relative dispersion may be quite different. In fact there is a most interesting regime of 'nonlocal transport' for relative dispersion before the asymptotic diffusion regime is reached. I think something should be commented on three different time regimes in relation with the very long times needed to observe consistently the diffusive regime for relative dispersion.

Following the reviewer's suggestion, we have added the following paragraph on p. 13 of the revised manuscript about the long times needed to observe diffusion in realistic

oceanic flows, and the different transport regimes that can occur on small to intermediate times: "The non-diffusive nature of the parcel motion over 90 days is because ocean eddies have finite length- and time-scales, so a variety of different transport regimes generally occurs before separating parcels become uncorrelated and transport becomes diffusive, as in a random walk. At very short times the motion of fluid parcels is largely governed by the local velocity shear, so the resulting transport regime is ballistic, i.e., $D\hat{a}\acute{L}\grave{I}T\hat{\ }2$ and $V\hat{a}\acute{L}\grave{I}T$ (Rypina and Pratt, 2017). At longer times, when velocity shear can no longer be assumed constant in space and time, the regime may transition to a local Richardson regime (i.e., $D\hat{a}\acute{L}\grave{i}t\hat{\ }3$), where separation at a given scale is governed by the local features of a comparable scale (Richardson 1926; Bennett 1984; Beron-Vera and LaCasce 2016), or to a non-local chaotic-advection spreading regime (i.e., $D\hat{a}\acute{L}\grave{I}exp\hat{a}\c{A}\c{a}(\lambda t)$), where separation is governed by the large scale flow features (Bennett 1984; Rypina et al. 2010; Beron-Vera and LaCasce 2016). The kinetic energy spectrum of a flow indicates whether a local or non-local regime will be relevant. The chaotic transport regime is generally expected to occur in mesoscale-dominated eddying flows, such as, for example, AVISO velocity fields, over time scales of a few eddy winding times. At times long enough for particles to sample many different flow features, such as Gulf Stream meanders or mesoscale eddies in the AVISO fields, the velocities of the neighboring particles become completely uncorrelated, and transport finally approaches the diffusive regime. With the mesoscale eddy turnover time being on the order of several weeks, it often takes longer than 90 days to reach the diffusive regime."

R2-6. Please use consistently either \delta x or L in the developments of Sect. 2.

Fixed, we now consistently use $\Delta$x.

R2-7. The quality of the figures is quite low, which is especially important for Fig. 3.

The quality of the figures has been improved in the revised manuscript.

Please also note the supplement to this comment:
https://www.nonlin-processes-geophys-discuss.net/npg-2017-63/npg-2017-63-AC1-supplement.pdf
* * *

---

## Author Comment (AC2) · 30 Jan 2018

The comment was uploaded in the form of a supplement:
https://www.nonlin-processes-geophys-discuss.net/npg-2017-63/npg-2017-63-AC2-supplement.pdf